# Socio-spatial disparities in access to emergency health care—A Scandinavian case study

**Jacob Hassler**⬤\*, **Vania Ceccato**

Department of Urban Planning and Environment, KTH Royal Institute of Technology, Stockholm, Sweden

\* jahass@kth.se

**Data Availability Statement:** All relevant data are within the manuscript and its Supporting information files.

**Funding:** VC recieved funding for this article from FORMAS (project number: 2016-00332).

## Abstract

Having timely access to emergency health care (EHC) depends largely on where you live. In this Scandinavian case study, we investigate how accessibility to EHC varies spatially in order to reveal potential socio-spatial disparities in access. Distinct measures of EHC accessibility were calculated for southern Sweden in a network analysis using a Geographical Information System (GIS) based on data from 2018. An ANOVA test was carried out to investigate how accessibility vary for different measures between urban and rural areas, and negative binominal regression modelling was then carried out to assess potential disparities in accessibility between socioeconomic and demographic groups. Areas with high shares of older adults show poor access to EHC, especially those in the most remote, rural areas. However, rurality alone does not preclude poor access to EHC. Education, income and proximity to ambulance stations were also associated with EHC accessibility, but not always in expected ways. Despite indications of a well-functioning EHC, with most areas served within one hour, socio-spatial disparities in access to EHC were detected both between places and population groups.

## Introduction

Where one lives determines the chances of one receiving an ambulance in adequate time. As ambulance travel times largely depends on the geographical distance between the patient and the closest ambulance, stations are usually located to supply as large areas as possible in as short time as possible. However, people are not equally served by ambulances. Rural populations tend to have longer wait times for ambulances [1,2], which in turn may have negative effects on health outcome and survival rates [2–4]. Assuring that the health care system can provide timely and adequate medical interventions despite varying geographical distances is therefore a key objective in emergency health care (EHC) planning, and an important aspect of social equity [5].

Considering demographic and socioeconomic characteristics of populations vary between urban and rural areas [e.g. 6,7] it is plausible that disparities in access to EHC exists between different segments of the population. Furthermore, in some emergency conditions, e.g. stroke,

Webpage: https://formas.se/en/start-page.html The
funders had no role in study design, data collection
and analysis, decision to publish, or preparation of
the manuscript.

**Competing interests:** The authors have declared
that no competing interests exist.

patients may require medical interventions at the emergency department [8]. Medical treatment in such cases is delayed not only by the response time of ambulances, but also by the trip to the emergency department. There may in other words be a difference of being reached by the ambulance and receiving medical intervention(s).

In this article, we attempt to reveal potential socio-spatial disparities in access to EHC in a Scandinavian case study. First, spatial variations in accessibility to EHC was assessed using a geographical information system (GIS). Several measures of accessibility were calculated to reflect scenarios when the patient needs to be taken to the hospital or not. These were 1) the time it takes for patients to be reached by the ambulance, i.e. Response Time (RT), then 2) the time it takes for patients to be transported to the hospital, i.e. the Transportation Time (TT), and finally, 3) the total time elapsed from when a call to the emergency services is made to when a patient arrives at the emergency department, i.e. the Total Prehospital Time (TPT). Then, we investigate how socio-spatial disparities in access to EHC may vary based on how 'accessibility' is measured. Lastly, regression modelling was carried out to assess whether some places, and population groups, are more likely to have poor levels of access.

The novelty of this study is two-fold. First, the study explicitly compares measures of accessibility and discuss implications that potential differences may have for planning purposes. This means that just because one can be reached quickly by an ambulance (RT), it does not necessarily mean that one can be quickly transported to the hospital (TT). The second novelty is that it identifies groups in society that have poor accessibility, and population based disparities in accessibility. Therefore, this study fills a gap in knowledge by showing how unequal access to EHC vary between socioeconomic and demographic groups, depending on which quantitative measure is used.

The article is structured as follows; first, a theoretical background and a literature overview on EHC accessibility and inequities in accessibility is presented. This is followed by a presentation of the case study. Then, the methods employed and the data that was used is presented, and the article concludes with a presentation of the results and a discussion, along with some conclusions that can be drawn from our study.

## Theoretical background and previous studies

### Inequitiable accessibility to emergency health care

Socio-spatial disparities in access to EHC have been argued to be driven by urbanization processes and population ageing, because such processes incentivize centralization of health care services in order to meet rising demand in cities [9,10]. When health care services are centralized and relocated from rural areas, the geographical distance between health care providers and rural areas often increase, leading to decreased levels of accessibility for rural populations. This is problematic considering rural populations are in many countries composed of increasing shares of older adults with relatively high demand for EHC [11]. As such, equitable provision of EHC is likely to become an increasingly important aspect of planning [9,12]. Improving accessibility in rural areas is therefore crucial in order to reduce socio-spatial disparities in access to EHC between urban and rural populations, and to reduce mortality rates following emergencies in rural areas [13]. This could be achieved for example by relocating specialist competence to rural areas [9]. However, possibilities to spread out health care services geographically is ultimately restricted by financial costs [14] which forces prioritization of some places and populations over others. Planning for equitable access to EHC is thus a major challenge. Below, a brief presentation of the concept of accessibility that underlie the spatial analysis performed in this study is presented.

## Spatial accessibility in health care

'Access' has been argued to be a measure of the "fit" between the supplier and the patient [15] and as ". . .a measure of potential and actual entry of a given population group into the health care system." [16]. Similar concepts of access, dependent on characteristics of both the user and the supplier of health-related services, have been formulated by others [15–18]. To enable the analysis of spatially variable levels of access to health care, 'spatial accessibility', relating to travel time and geographic distance between the user and the supplier of services, has been developed [19].

Previous studies on socio-spatial disparities in access to EHC have often calculated the time elapsed between calling an ambulance and arriving at the hospital, i.e. TPT, as a measure of accessibility [e.g. 6,20,21]. However, as geographical contexts vary largely between countries, regions and cities, it is problematic to employ universal thresholds of what constitutes high or low accessibility. Several studies [e.g. 6,20,22] have therefore employed 'the golden hour' concept to determine whether an area or a group have poor levels of access to EHC. The concept refers to when a patient can reach the hospital within one hour from calling the emergency services, i.e. having a TPT below one hour, a universal benchmark for 'good' access to EHC.

However, the way that access is measured may influence the spatial patterns of service [23]. Ambulance response times (RT) are traditionally reported in EHC research and policy [12], but calculating the time thresholds that make up the time elapsed from when a patient contacts the emergency services to when he or she receives medical intervention at the emergency department (TPT) may more accurately account for confounding factors such as the distance from the emergency scene and the nearest hospital, which influence the transportation time (TT) (ibid). As the timeliness in which EHC is provided influence the outcome health status of patients in emergency conditions [e.g. 24,25], and that some emergencies can be treated on site while others need interventions at the emergency department [8], it is important to measure not only ambulance RT but also the TT and the TPT. Below, a summary of some previous research that measure accessibility using these measures are presented.

## Previous studies on socio-spatial disparities in access to EHC

Accessibility to EHC tends to be lower in rural areas than urban areas [e.g. 1–4,6,22,23,26–28], and mortality rates in the prehospital setting following emergency conditions such as trauma [29], cardiac arrest [2] and stroke [30] has been shown to be higher for patients in rural areas than in urban areas. Disparities in levels of accessibility has also been observed between demographic and socioeconomic groups. For example, in the urban setting, foreign born individuals [31] and populations living in materially deprived areas [32] have been shown to have poorer accessibility to trauma centers. Residential segregation may also facilitate inequitable levels of access to health care for minority groups [33,34] and between ethnic groups [31,35]. Likewise, low income levels [12,22,31,35], not having a medical insurance [22] and old age [6,35] has been related to low levels of accessibility.

Other factors may influence the likelihood of a patient calling the ambulance, and thus their level of accessibility to EHC. For example, lacking public knowledge of symptoms may lead to hesitation of bystanders to call the ambulance [35,36]. Self-reported life quality and loneliness or anxiety [37], having social support, e.g. family members [38], and individual's perceptions of the health care system [39] has also been shown to influence the likelihood of a patient calling the ambulance.

While most studies on access to EHC are situated in a Western context, socio-spatial disparities in access have been observed also in, for example, Brazil [40], Ghana [41] and in sub-Saharan Africa [42]. In Sweden, residents in areas with lower socioeconomic status have been

found to run a higher risk of suffering an out of hospital cardiac arrests [43]. However, like several other international studies have reported, e.g. in Korea [44] and Singapore [45], this relationship diminished with age, where older adults run the highest risk of suffering an out of hospital cardiac arrest [43]. Socioeconomic status has also been linked to survival rates in Sweden, where lower education and income levels were found to be predictive of low survival rates in 30 days following an out of hospital cardiac arrest [46]. Education has also been argued by others to be a primary explanatory variable for health disparities in the Swedish context [47].

## Research design

In this study we investigate potential socio-spatial disparities in access to EHC. As some time critical emergency conditions require medical interventions in the hospital [29,30] rather than on-site, we suggest that access to EHC can be divided into three parts to reflect accessibility for conditions that require different types of treatment. These are the time elapsed from when a patient calls emergency services to when an ambulance arrives at the emergency scene (RT), the time elapsed to transport the patient from the scene to the emergency department (TT) and the total time elapsed from when a call to the emergency services is made to when the patient arrives at the emergency department (TPT). Such division of measures may indicate accessibility for various types of emergencies and has been motivated previously as being more accurate than using solely response times, as it accounts for potential confounding factors [12] and distance from definitive care [12,23]. Drawing on previous research [9,12], we suggest that disparities in access to EHC between urban and rural areas may be exacerbated by population characteristics and changing demographic structures, particularly ageing.

Considering Sweden is undergoing demographic changes related to urbanization processes and ageing of the population, especially in rural areas, as well as health care reforms that affect the ability to provide EHC [5], it is important to assess population based disparities in access to EHC. Therefore, in this study, we aim to answer the following research questions:

- How does accessibility to EHC vary spatially?

- Are there differences in accessibility depending on how it is measured?

- How does disparities in accessibility vary between socioeconomic-demographic population groups?

Below we introduce the study area and discuss step by step the procedures involved in the execution of the study, from data acquisition to modelling.

## The study area

Sweden is one of Europe's largest country and has the 4th lowest population density of the EU countries [48] with a total population of roughly 10 million people [49]. In 2017, Sweden allocated 11% of the national GDP on health care, the third highest in the EU [50]. The health care system is to a large degree publicly funded, i.e. 84% of total expenditures (ibid.) and administrative regions are responsible for providing EHC to anyone that require it [51]. Regions are free to outsource ambulance operations fully or partly to private contractors (ibid.). In 2018, a total of 3,196,642 calls were made to emergency services in Sweden [52] and the median response time for ambulances was 15 minutes nationally, varying regionally between 12 and 22 minutes [53].

While Sweden is internationally regarded as a well-functioning and relatively equitable country in most regards it constitutes an interesting case study because the reorganization of EHC in the country follows an international trend [14], where dwindling population numbers

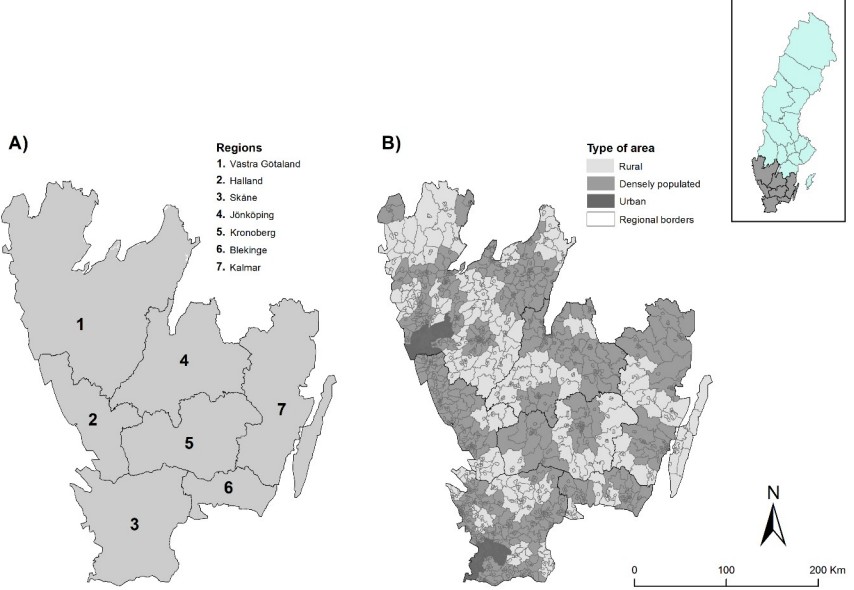

**Fig 1. The study area: Southern Sweden.** Overview map of the study area. Source: Authors.

in rural areas in the last decades has in many parts of the country led to a relocation of specialist services, and as a result also of many emergency departments [5]. Around 40% of emergency departments have been closed or relocated in Sweden since the 1970's [51], which has indirectly reduced the ability to deliver EHC services in rural areas because specialist expertise is often required to treat emergency patients [54]. Currently, there is a lack of knowledge on how EHC accessibility varies on the macro level following such changes [5] because the EHC system is planned, and measured, on a regional level [55]. Also, current policy goals related to EHC accessibility in Sweden focus solely on reaching a certain share of the population within a set time threshold and makes no differentiation between either urban and rural areas, nor between population groups [56].

Southern Sweden is visualized in Fig 1 including (A) administrative regions and (B) the units of analysis, i.e. demographic statistical areas (DeSO), classified by rural, densely populated and urban areas. Southern Sweden was selected as study area because it has (a) a mix of urban and rural municipalities; (b) a population density that is geographically more homogenous than the rest of the country and (c) because of data availability due to the fact that locations of ambulance stations in Sweden are currently not available in a single database. About 4.3 million people, or about 40% of Sweden's total population of roughly 10 million people [49], live in the study area.

The study area is composed of *urban municipalities*, which contain less than 20% of the population living in rural areas and, combined with neighboring municipalities, have more than 500,000 residents. *Densely populated municipalities*, which have less than 50% of the population living in rural areas, where at least half of the population has commute times less than 45 minutes to cities with 50,000 or more residents, and finally, *rural municipalities* with at least 50% of the population living in rural areas, where at least half of the population has commute times less than 45 minutes to cities with 50,000 or more residents [57]. Roughly 40% of the municipalities in the study area are considered rural, while 8% are categorized as urban, and roughly 50% as densely populated.

## Data and methods

### Data

The study was spatially delimited by administrative borders on several geographical levels. The units of analysis were DeSO areas, provided by Statistics Sweden (SCB) [58], a Swedish system of spatial division based on areas that contain around 1,000 to 2000 persons (ibid.). Data on population numbers in a grid of 1 km by 1 km squares was downloaded from Statistics Sweden [59] to calculate the point in each DeSO area where the largest share of the population lived in order to weight centroids of areas similar to how others have done before us [e.g. 21]. Socio-economic and demographic data was downloaded from Statistics Sweden [60]. The characteristics of the data and spatial divisions are presented in S1 Table.

The Swedish national road network database (NVDB) [61] was imported to a GIS to calculate travel times. Hospitals with emergency departments (N = 25) were extracted manually from Google Earth based on the names of hospitals within the study area provided by the National Board of Health and Welfare [62]. As no official registry of ambulance stations in Sweden exists, these were gathered from each region individually. Out of a total of 118 stations, five were not retrievable by exact address and were thus placed arbitrarily in the center of the city or town they were located in, before being imported to a GIS.

### Methods

**Analysis of spatial accessibility to EHC.** The road network was imported to ArcGIS, and travel times for each road segment was calculated by dividing the length of roads with the speed limits. Then, ambulance stations, hospitals and the DeSO areas were imported. To control for border effects, where areas close to the border of the study are could potentially be reached faster from ambulances dispatched from other regions, we also included stations, hospitals and roads from neighboring regions in the north. Ambulance stations and hospitals were relocated to the closest junction (where two road segments connect) to assure that drive times would always be calculated by traversing entire lines. We also weighted the centroids of the DeSO areas by population numbers obtained from Statistics Sweden [59] to improve the accuracy of the analysis. For areas where population data was not available, the centroids were included without weighting.

To calculate accessibility measures, we ran a network analysis (Closest Facility) in ArcGIS. Response times (RT) were calculated as the travel time between ambulance stations and each individual DeSO area, with added activation times adopted from Carr et al. [24]. Transportation times (TT) were calculated in the same manner, but strictly as the travel time between emergency departments and patients.

RT was calculated for each individual area by converting the DeSO areas from polygons to centroids, and then importing them as 'incidents' in the network analysis. The ambulance stations were imported as 'facilities', representing the points of departure for ambulances. Individual routes containing information about travel times were generated for each centroid. To account for the time elapsed between receiving a call and dispatching an ambulance, a value of 1.4 minutes was added to all urban and densely populated areas and 2.9 minutes was added to rural areas, based on the activation times suggested by Carr et al. [24]. Then, we changed the 'facilities' from ambulance stations to emergency departments to calculate TT.

Lastly, Total prehospital time (TPT) was calculated by combining RT and TT and adding on-scene times adopted from Carr et al. [24]. The on-scene times were assigned based on which type an area had been assigned: 13.5 minutes in urban and densely populated areas and 15.1 minutes in rural areas (ibid.).

To make our results comparable with international research [e.g. 6,20,22] we employed the concept of the 'golden hour' as a threshold for assessing accessibility. Although this concept is a widely adopted term in research and trauma system planning, it has been criticized for not being supported scientific and medical research [63] and the benefits of planning trauma systems according to the 'golden hour' concept has been increasingly questioned in research [64]. Regardless, it provides a commonly agreed upon benchmark for comparing international levels of accessibility, making it suitable for the purpose of this study, i.e. to make the results comparable.

## Assessment of differences in accessibility by measures

Following the network analysis, we tested the relationship between RT and TT to find out whether levels of accessibility were different depending on which measure is used. A Pearson correlation test was carried out for this purpose, which indicated a statistically significant correlation between the two measures.

To understand whether, and to what degree, the correlation between the accessibility measures varied spatially, we then ran an analysis of variance (ANOVA) test where the variance of each measure was separated by urban, densely populated and rural areas. The test indicated significant differences in variation between the types of areas for both measures. To find out where the differences occurred, a Tukey post-hoc test was carried out.

## Modelling disparities in accessibility

Similar to previous research [e.g. 12,22,65] we carried out a regression analysis to assess if levels of access to EHC varied along socio-spatial lines. The dependent variables were the measures calculated in the network analysis, i.e. RT, TT and TPT. All measures exhibited strong right-tail skewed structures, and persisting non-normal data distribution despite being logarithmically transformed, as well as being overdispersed. Following common procedure when facing these problems [66], and like other studies that has modelled count data in studies on EHC accessibility [12], we opted for a negative binominal regression model.

The selection of covariates were based on the literature. A bivariate correlation test indicated that some variables were highly correlated (Pearson > = 0.6), see S2 Table, which led us to drop some candidate variables. The predictor variables included in the analysis were thus; the shares of older adults, shares of highly educated and yearly median income levels. We also included dummy variables that indicated if a municipality had an ambulance station or not, if an area was located in a rural, densely populated or urban municipality and for each administrative region to account for potential regional differences. Knowing that rural areas tend to have relatively large shares of older adults in Sweden [67,68] and internationally [7,69], we added an interaction between the rural dummy variable and the share of older adults to assess whether older populations in rural areas had specifically poor levels of access.

Lastly, to the author's knowledge, no universal standard similar to the 'golden hour' exists for RT or TT. To avoid using time thresholds that could be inappropriate in the Swedish context when analyzing socio-spatial disparities in access to EHC, we split the measures into quintiles where the 5th quintile represents the most underserviced areas. This allowed us to assess how the explanatory power of the predictor variables in relation to the measures of accessibility varied between the entire study area and in underserviced areas.

## Results

### Spatial patterns of accessibility to EHC

The results indicate that almost 90% (n = 2256) of the areas are within the 'golden hour', meaning that emergency patients in these areas have below one hour TPT. Meanwhile roughly 10%

**Table 1. Descriptive statistics for the socioeconomic, demographic and area type variables.**

| Variables | Entire study area | Within the golden hour (TPT below 1 hour) | Outside the golden hour (TPT above 1 hour) |
|---|---|---|---|
| *Demographic and socioeconomic variables* | Median | Median | Median |
| Highly educated—share of population* | 22% | 23,5% | 14,9% |
| Median income levels (SEK per year)* | 280,252 | 284,433 | 254,783 |
| Older adults—share of population* | 20,4% | 19,7% | 26% |
| Total population* | n = 4,331,255 | n = 3,923,545 | n = 407,710 |
| *Spatial variables* | | | |
| Area types** | | | |
| Rural | 19.7% (n = 497) | 14.8% (n = 334) | 61.5% (n = 163) |
| Densely populated | 52.4% (n = 1321) | 54.1% (n = 1220) | 38.5% (n = 102) |
| Urban | 27.9% (n = 702) | 31.1% (n = 702) | - |
| Total | n = 2520 | n = 2256 | n = 265 |

\* DeSo level.

\*\* Municipal level.

Descriptive statistics separated by the 'golden hour' threshold.

of the areas (n = 265) were found to have TPT's above one hour, where 163 out of 497 are rural areas and 102 out of 1321 are densely populated areas. No urban areas have TPT above one hour. The median share of highly educated and the median yearly income levels are lower in poorly serviced areas, with differences around 8% and 30 000 Swedish Crowns (SEK) respectively. In addition, the median share of older adults is roughly 6% higher in poorly serviced areas. Descriptive statistics for the socioeconomic, demographic and spatial variables are presented in Table 1, separated by the entire study area, and by areas within and outside of the 'golden hour'.

Mean RT for the entire study area was 9.2 minutes, while mean TT was 15.5 minutes and mean TPT was 38.5 minutes, see S3 Table. However, accessibility varied between urban, densely populated and rural areas. The largest variations were observed in rural areas, where RT ranged from 3.2 to 32.4 minutes, while TT ranged from 6 to 83.6 minutes and TPT ranged from 30.9 to 108.8 minutes. In densely populated areas, the measures ranged between 1.5 to 39.8 minutes (RT), 0.4 to 70 minutes (TT) and 15.7 to 100.7 minutes (TPT). Urban areas had the lowest variations, and the lowest wait times in general for all measures, where RT ranged from 1.6 to 17.8 minutes, TT from 0.35 to 25.4 minutes and TPT from 18.8 to 52.7 minutes.

The spatial patterns of EHC accessibility are visualized in Fig 2, where RT (A), TT (B) and TPT (C) are presented separately. The spatial patterns observed for RT and TT suggest that areas close to regional borders have relatively longer wait times both to be reached by and ambulance (RT) and to be transported to the hospital (TT), resulting in a high total prehospital time (TPT) in these areas. A total summary of RT, TT and TPT for each region can be seen in S3 Table.

## Assessing differences in accessibility measures

An ANOVA test showed that there are significant variations between urban, densely populated and rural areas for both RT and TT. These variations are different for RT and TT, meaning that while there are spatial variations in RT these are not as large as they are for TT. Spatial variations for RT is in other words not indicative of variations in TT. Mean values varied between types of area for both RT ($F(2, 2522) = 150.1$, $p < .01$) and TT ($F(2,2522) = 628,7$, $p < .01$), and a Tukey post-hoc test revealed that the mean time to reach patients (RT) in rural areas

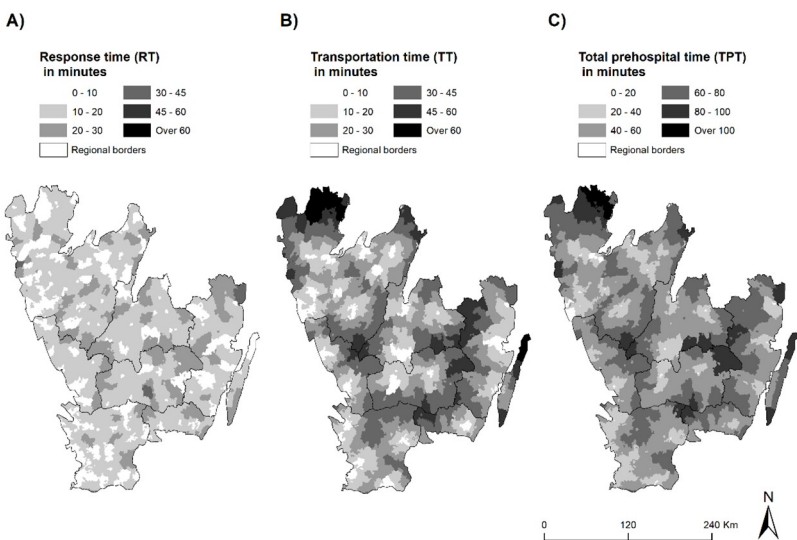

**Fig 2. Visualizations of accessibility measures.** Maps showing how RT, TT and TPT vary across the study area. Source: Authors.

(12.2 ± 6.1 min, p < .01) was statistically significantly longer than in densely populated areas (9.3 ± 5.6 min, p < .01) or urban areas (7.1 ± 2.8 min, p < .01). For TT, the relationships were similar: times were statistically significantly longer in rural areas (28.5 ± 10.8, p < .01) than in densely populated areas (14.5 ± 11.5, p < .01) or urban areas (7.98 ± 4.9, p < .01). The variance in mean values between groups was larger for TT than RT, see S1 Fig. In the following section, we turn to the results from the regression analysis on how access to EHC varied between population groups.

## Modelling disparities in accessibility to EHC

Three models were run for each separate measure (RT, TT and TPT) with the entire study area included (N = 2520), see Table 2. Higher shares of older adults were statistically significant and explained variation in access to EHC, regardless of modeling strategies and EHC measures. Likewise, areas with higher shares of highly educated was associated with lower wait times for all measures for almost all models. Unsurprisingly, having an ambulance station explained lower wait times for all measures.

We then added an interaction effect between older adults and rural areas (Table 2). Note that in all models without an interaction factor (rural x older adults), the dummy variable indicating rural areas was associated with longer wait times for all measures. In the models where the interaction effect was included, the rural dummy variable was rendered insignificant. Interestingly, both the share of older adults and the interaction factor (rural x older adults) turned out to be significant in all EHC models.

To identify the covariates of the areas that are poorly served, we also modelled only the observations in the 5th quintile (N = 504), i.e. the poorest served areas (Table 3). While higher shares of older adults significantly explained longer wait times in all measures in these models as well, the explanatory power of the other covariates varied between the measures. Having an ambulance station in one's municipality significantly explained lower RT while, surprisingly, significantly explaining longer TT. Higher median income levels explained lower TT, while higher shares of highly educated explained lower TPT.

**Table 2. Negative binominal regression modelling results for each accessibility measure, without (left) and with (right) interaction effect.**

| | Without interaction | | | | | | With interaction | | | | | |
| | RT | | TT | | TPT | | RT | | TT | | TPT | |
| Variables | Coef. | Sig. | Coef. | Sig. | Coef. | Sig. | Coef. | Sig. | Coef. | Sig. | Coef. | Sig. |
|---|---|---|---|---|---|---|---|---|---|---|---|---|
| Older adults | 1.006*** | 0.000 | 2.333*** | 0.000 | 1.081*** | 0.000 | 0.819*** | 0.000 | 1.926*** | 0.000 | 0.836*** | 0.000 |
| High education | -0.889*** | 0.000 | -3.222*** | 0.000 | -1.192*** | 0.000 | -0.920*** | 0.000 | -3.287*** | 0.000 | -1.231*** | 0.000 |
| Median income | 0.000*** | 0.000 | 0.000*** | 0.000 | 0.000*** | 0.000 | 0.000*** | 0.000 | 0.000*** | 0.000 | 0.000*** | 0.000 |
| Ambulance station | -0.444*** | 0.000 | -0.210*** | 0.000 | -0.186*** | 0.000 | -0.450*** | 0.000 | -0.219*** | 0.000 | -0.193*** | 0.000 |
| Urban (ref) | 0.000 | | 0.000 | | 0.000 | | 0.000 | | 0.000 | | 0.000 | |
| Densely populated | 0.062* | 0.050 | -0.049 | 0.200 | -0.002 | 0.930 | 0.067* | 0.030 | -0.038 | 0.320 | 0.005 | 0.790 |
| Rural | 0.250*** | 0.000 | 0.429*** | 0.000 | 0.302*** | 0.000 | -0.004 | 0.970 | -0.089 | 0.480 | -0.025 | 0.680 |
| Region Blekinge (ref) | 0.000 | | 0.000 | | 0.000 | | 0.000 | | 0.000 | | 0.000 | |
| Region Halland | -0.126* | 0.040 | -0.152* | 0.040 | -0.092** | 0.010 | -0.128* | 0.030 | -0.161* | 0.020 | -0.095** | 0.010 |
| Region Jönköping | -0.090 | 0.130 | -0.217** | 0.000 | -0.095** | 0.000 | -0.087 | 0.140 | -0.216** | 0.000 | -0.093** | 0.010 |
| Region Kalmar | -0.100 | 0.100 | -0.246*** | 0.000 | -0.107** | 0.000 | -0.106 | 0.080 | -0.269*** | 0.000 | -0.119*** | 0.000 |
| Region Kronoberg | -0.007 | 0.920 | -0.218** | 0.010 | -0.085* | 0.020 | -0.010 | 0.880 | -0.231** | 0.000 | -0.090* | 0.020 |
| Region Skåne | -0.111* | 0.030 | -0.305*** | 0.000 | -0.148*** | 0.000 | -0.114* | 0.030 | -0.317*** | 0.000 | -0.153*** | 0.000 |
| Region Västra Götaland | 0.260*** | 0.000 | -0.309*** | 0.000 | -0.177*** | 0.000 | -0.265*** | 0.000 | -0.325*** | 0.000 | -0.184*** | 0.000 |
| Interaction: Rural x Older adults | - | - | - | - | - | - | 1.066* | 0.010 | 2.182*** | 0.000 | 1.377*** | 0.000 |
| Intercept | 2.263*** | 0.000 | 2.547*** | 0.000 | 3.675*** | 0.000 | 2.300*** | 0.000 | 2.628*** | 0.000 | 3.722*** | 0.000 |
| N | 2520 | | 2520 | | 2520 | | 2520 | | 2520 | | 2520 | |
| Log likelihood | 7124 | | 8492 | | 9370 | | 7121 | | 8482 | | 9352 | |
| BIC | 14358 | | 17095 | | 18850 | | 14359 | | 17083 | | 18822 | |
| AIC | 14277 | | 17013 | | 18768 | | 14272 | | 16995 | | 18735 | |

* p<0.05,

** p<0.01,

*** p<0.001.

In the models including the entire study area (Table 2), the significant covariates were largely similar for all measures. The regional dummy variables were, however, less powerful in explaining variations in RT than TT and TPT. In the models including only the poorly served areas (Table 3) variance in the measures was explained by different covariates, although older adults explained variations in all measures in these models too. Model diagnostics (AIC, BIC and log likelihood values) indicated lower values for the RT models than the TT and TPT models, suggesting that the goodness of fit was best for RT.

## Discussion

Around 90% of the areas (3,923,545 individuals) in southern Sweden have access to EHC at the emergency department within the 'golden hour'. Internationally this region performs well, as timely access to EHC using road or air ambulance has been observed for 88% of the US population [20,22], 83% of the population in New Zeeland [6] and 77% of the population in Canada [28]. In Scotland, 94% of the population have been argued to have access to EHC within 45 minutes [27]. Unlike these studies, our study was carried out only in southern Sweden rather than on the national level. A national study of accessibility to EHC in Sweden would likely indicate lower accessibility as the study area's population density is much higher (107 inhabitants per $km^2$) than in the whole country (20.5 per $km^2$).

Table 3. Negative binominal regression modelling results for the poorest serviced areas.

| | Poorly serviced areas (5th quintile) | | | | | |
|---|---|---|---|---|---|---|
| | RT | | TT | | TPT | |
| Variables | Coef. | Sig. | Coef. | Sig. | Coef. | Sig. |
| Older adults | 1.225*** | 0.000 | 0.649** | 0.002 | 0.742*** | 0.000 |
| High education | -0.302 | 0.143 | -0.248 | 0.213 | -0.325** | 0.008 |
| Median income | 0.000 | 0.772 | -0.002*** | 0.000 | -0.001 | 0.070 |
| Ambulance station | -0.099*** | 0.001 | 0.103** | 0.003 | 0.005 | 0.792 |
| Urban (ref) | 0.000 | | 0.000 | | 0.000 | |
| Densely populated | -0.014 | 0.860 | 0.241 | 0.100 | 0.020 | 0.830 |
| Rural | -0.040 | 0.617 | -0.190 | 0.195 | -0.028 | 0.759 |
| Region Blekinge (ref) | 0.000 | | 0.000 | | 0.000 | |
| Region Halland | -0.059 | 0.279 | -0.038 | 0.499 | 0.034 | 0.329 |
| Region Jönköping | -0.000 | 0.997 | 0.001 | 0.983 | 0.008 | 0.772 |
| Region Kalmar | 0.012 | 0.830 | 0.048 | 0.335 | 0.018 | 0.536 |
| Region Kronoberg | 0.046 | 0.410 | 0.010 | 0.853 | 0.038 | 0.231 |
| Region Skåne | -0.106* | 0.025 | -0.088 | 0.053 | -0.031 | 0.260 |
| Region Västra Götaland | -0.046 | 0.317 | 0.082 | 0.055 | 0.074** | 0.004 |
| Intercept | 2.747*** | 0.000 | 3.748*** | 0.000 | 4.162*** | 0.000 |
| N | 504 | | 504 | | 504 | |
| Log likelihood | -1344 | | -1729 | | -1797 | |
| BIC | 2769 | | 3546 | | 3682 | |
| AIC | 2714.3 | | 3487.3 | | 3622.6 | |

* $p < 0.05$,

** $p < 0.01$,

*** $p < 0.001$.

Meanwhile, around 10% of the areas (407,710 individuals) do not have access to EHC within the 'golden hour'. Similar to findings in previous studies [6,20,22,27] a majority of these live in rural areas, indicating a rural-urban divide in accessibility. Like Hsia & Shen [12] observed in the US, approximately one third of rural areas had poor accessibility to EHC in southern Sweden. Rurality also explained poorer access in the global regression models (Table 2) which is in line with findings from previous studies [1–4,6,22,26].

Rurality alone may however not necessarily explain variations in access to EHC. Regression modelling showed that rural areas with high shares of older adults have particularly poor levels of access to EHC. While areas with high shares of older adults have relatively poor levels of access to EHC in general, as has been observed elsewhere [6,35], the interaction factor based on the 'older adults' variable and the rural dummy variable rendered the rural dummy variable insignificant, indicating that rural areas with high shares of older adults have particularly poor accessibility to EHC.

This is an important finding because older adults has been shown to run a higher risk of suffering emergency conditions such as cardiac arrests internationally [44] as well as in Sweden [43]. Likewise, education levels has been argued to be a key factor in explaining varying health status in the Swedish context [47]. Considering that highly educated populations had relatively good access while older adults had relatively poor levels of access, access to EHC may be poor for those who need it the most, i.e. older adults, and better for those that have lower health care needs, i.e. highly educated. Recognizing such variations in accessibility is crucial for future planning of equitable EHC provision, a key objective for the Swedish health care system [47].

Furthermore, our results indicate that areas located in municipalities that have an ambulance station do not necessarily have better access to EHC. In contrast, when looking only at the most poorly served areas (Table 3), this variable explained longer TT. This was unexpected because intuitively, the geographic relation between patients and ambulance stations should not influence TT. Likely, this depends both on the fact that ambulance stations are often localized to remote areas to maximize coverage, and also on the fact that only the poorest served areas were included in the model. Areas in close proximity to an ambulance station naturally have relatively low RT. However, the remote areas where stations were localized would be expected to have relatively high TT's compared to the more centrally located suburbs and densely populated areas. This explains why ambulance stations indicated higher TT, and lower RT, among the poorest served areas. Similarly, another unexpected finding was that higher median income levels explained poorer accessibility in the entire study area, opposite to what several others have reported [12,22,31], while only significantly explaining lower TT in the models where only the poorest served areas were included. This could be explained by the fact that income levels tend to be relatively high in residential suburbs and outskirts of cities. Compared to the most remote areas, i.e. the poorest served areas, travel times to the emergency departments that are generally located in cities would expectedly be lower here.

We found differences between the measures in terms of which variables were explanatory of variations in RT, TT and TPT in the models with only the poorly served areas. This indicates that the population groups that have poor access to EHC are not necessarily the same in urban, densely populated and rural areas, which has been observed also by others [e.g. 31]. It is an important observation because conclusions drawn from studies situated in urban areas may not necessarily hold true in rural areas and vice versa. Therefore, we argue that accessibility to EHC needs be studied on various scales and with different spatial divisions to avoid drawing general conclusions that do not hold true in some areas, and to minimize the effect of the modifiable areal unit problem. This may be particularly true in the Swedish context where differences in levels of access between urban and rural areas is currently not reflected in policy goals [56].

Several other variables could be employed to study disparities in accessibility, e.g. mental health status [37] and perceptions of health care [39]. It is plausible that the inclusion of such factors would indicate specific groups that are less likely to call the ambulance. Regardless of whether those groups or individuals could access EHC quickly, if they do not call the ambulance in the first place their 'actual' accessibility would be low. The disparities in levels of accessibility observed here could likely be broken down further by including such variables.

To sum up, socio-spatial disparities in accessibility to EHC poses a challenge for future planning of equitable emergency health care systems, especially in the light of ongoing centralization processes that tend to limit the ability to provide timely EHC in rural areas, and thus may facilitate increased disparities in accessibility.

To decrease the number of areas with low levels of accessibility, and to make access equitable for all parts of the population, we suggest that health care policymakers and governmental bodies should employ not only response times (RT), but also the transportation times (TT) to the hospital and the total prehospital time (TPT) when planning ambulance services. Focusing solely on RT in policy goals risks obscuring poor accessibility for patients that require medical interventions in the hospital, especially for populations in rural areas.

## Limitations

This study has a number of limitations. We only analyzed road based ambulances, unlike several others who include air transportation when calculating accessibility to EHC [e.g. 6,22]. This choice was motivated by two primary reasons; that ambulance helicopters in Sweden in

many cases cannot be dispatched due to poor weather conditions [70] and because only one of the regions in the study area have access to ambulance helicopters (ibid.).

In addition, we assume that an ambulance is always ready to be dispatched from a station. In reality, they are often dispatched directly from another mission, especially in emergency situations, and sometimes an ambulance may not be available. Response times may also be delayed by factors such as road conditions, weather or traffic. Omitting such factors in the analysis, it is plausible that we overestimate accessibility as such factors would likely increase travel times. Another limitation is that the position of DeSO area centroids are less accurate in rural areas because the DeSO system is based on population density. Thus, decreased population density entails larger geographic areas. We also assume that all emergency departments are capable of treating emergency patients which is not always the case.

Lastly, we employ the 'golden hour' concept to compare our results with other studies. While the concept has been criticized for not being rooted in medical science [63] and thus not being theoretically sound to base EHC planning on, it does allow for comparison. For our purpose, i.e. to compare disparities in accessibility with other countries, it is useful and, to the best of the author's knowledge, it is the only widely adopted time threshold that can be used for this purpose in an international setting.

## Conclusions

In this study, we investigated socio-spatial disparities in access to emergency health care in southern Sweden. Spatial disparities in accessibility were found between urban and rural areas, where rural areas had poorer levels of accessibility in general. However, these disparities varied depending on whether a patient needed to be transported to a hospital to receive medical interventions or not, and larger spatial disparities was observed for transportation time (TT) than for response times (RT). This is important because physical relocation of emergency departments and ambulance stations inevitably influence travel times between patients and EHC suppliers, and longer TT may be obscured by low RT. In essence, short RT does not necessarily entail short TT or TPT. Disparities were also observed between population groups. Education levels and income levels, and the share of older adults, explained variations in accessibility. Rural areas with high shares of older adults were found to be particularly underserviced. The results presented here highlights how social and spatial disparities in levels of accessibility are often intertwined.

## Suggestions for future studies

Future studies should therefore build on the knowledge of equitable access to EHC, for example by including temporal aspects in the analysis to see how accessibility changes throughout the day. The method employed here should also be applied to analyze accessibility on the national level in Sweden. We also suggest future studies incorporate a wider range of factors simultaneously in the analysis to expand the knowledge on disparities in accessibility to EHC.

## Supporting information

**S1 Fig. Boxplots of response time (RT) and transportation time (TT) grouped by type of area, 2018.** Source: Authors.
(TIF)

**S1 Table. Characteristics of the dataset.**
(PDF)

**S2 Table. Bivariate correlation matrix.** $^*$ p<0.05.
(PDF)

**S3 Table. Descriptive statistics of RT, TT and TPT for different types of areas and for different administrative regions.** $^*$All values in minutes.
(PDF)

## Acknowledgments

We would like to acknowledge the administrative regions that supplied us with information of the location of ambulance stations, without which we would not have been able to carry out this study. These regions were Region Kalmar län, Region Jönköpings län, Region Halland, Region Skåne, Region Kronoberg and Västra Götalandsregionen.

## Author Contributions

**Conceptualization:** Jacob Hassler, Vania Ceccato.

**Data curation:** Jacob Hassler.

**Formal analysis:** Jacob Hassler, Vania Ceccato.

**Funding acquisition:** Vania Ceccato.

**Investigation:** Jacob Hassler, Vania Ceccato.

**Methodology:** Jacob Hassler, Vania Ceccato.

**Project administration:** Vania Ceccato.

**Resources:** Vania Ceccato.

**Software:** Jacob Hassler.

**Supervision:** Vania Ceccato.

**Validation:** Vania Ceccato.

**Visualization:** Jacob Hassler.

**Writing – original draft:** Jacob Hassler.

**Writing – review & editing:** Vania Ceccato.

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
