## [Decision Letter · Decision Letter 0]

27 Sep 2021

PONE-D-21-22117Socio-spatial disparities in access to emergency health care - a Scandinavian case studyPLOS ONE

Dear Dr. Hassler,

Thank you for submitting your manuscript to PLOS ONE. After careful consideration, we feel that it has merit but does not fully meet PLOS ONE’s publication criteria as it currently stands. Therefore, we invite you to submit a revised version of the manuscript that addresses the points raised during the review process.

ACADEMIC EDITOR: Considering my own reading and reviewer suggestions, I am recommending a major revision for this paper.  Looking forward to see a revised version. 

We look forward to receiving your revised manuscript.

Kind regards,

Srinivas Goli, Ph.D.

Academic Editor

PLOS ONE

Journal Requirements:

2. We note that Figures 2 & 3 in your submission contain map images which may be copyrighted. All PLOS content is published under the Creative Commons Attribution License (CC BY 4.0), which means that the manuscript, images, and Supporting Information files will be freely available online, and any third party is permitted to access, download, copy, distribute, and use these materials in any way, even commercially, with proper attribution. For these reasons, we cannot publish previously copyrighted maps or satellite images created using proprietary data, such as Google software (Google Maps, Street View, and Earth). For more information, see our copyright guidelines: http://journals.plos.org/plosone/s/licenses-and-copyright.

a. You may seek permission from the original copyright holder of Figures 2 & 3 to publish the content specifically under the CC BY 4.0 license.  

Additional Editor Comments:

Considering my own reading and reviewer suggestions, I am recommending a major revision for this paper. Looking forward to see a revised version.

Reviewers' comments:

Reviewer's Responses to Questions

**Comments to the Author**

1. Is the manuscript technically sound, and do the data support the conclusions?

Reviewer #1: Partly

Reviewer #2: Yes

Reviewer #3: Yes

2. Has the statistical analysis been performed appropriately and rigorously? 

Reviewer #1: I Don't Know

Reviewer #2: Yes

Reviewer #3: Yes

3. Have the authors made all data underlying the findings in their manuscript fully available?

Reviewer #1: No

Reviewer #2: Yes

Reviewer #3: Yes

4. Is the manuscript presented in an intelligible fashion and written in standard English?

Reviewer #1: No

Reviewer #2: Yes

Reviewer #3: Yes

5. Review Comments to the Author

Reviewer #1: Dear Respectable Authors

Thank you for considering a great area of research related to socio-capital disparities and its impact on access to emergency health care. Your manuscript is well-design but the structure if your manuscript needs revision. Please look at the similar article and re-structure your manuscript. I recommend some revision for better reporting the manuscript as follow;

- Abstract, please add statistical analysis in brief in this section.

- The whole structure of your manuscript is not good and for example, the introduction and the methods sections are a bit messy. Some information regarding method reported in the introduction. Please report the method based on the stages of the study. Also the results based on the study questions.

- Please separate reference, for example references 1-5. Also, in my opinion the number of references is high and some references not closely related to your topic. for example references 2 and 4.

- Please add the gap of evidence and also the novelty of your work in the introduction section.

- Provide a brief description of the emergency care system in the country in terms of statistics.

- In general, the article is very long. There is no need to review texts with this extent and it is better to summarize.

- lines 178-9, on what basis are these suggestions made?

- The quality of figures is low. Please remove Fig 1.

- In question 3, you stated some references. Explain the reason.

- In my opinion you have 5 questions not 3.

- The evidence related to "golden hour" is not enough at both national and international levels.

- SEK? please add full term for the first time.

- Please report demographic information regarding table one at the first paragraph of the results section.

- Please separate the conclusion and the recommendations. IN conclusion you needs briefly address the answer to the aim/question of the study regardless of statistical difficulties. Also, remove recommendation from conclusion and report it under separate subheading.

- what is your practical recommendation for health policymaker and government bodies to decrease the number of state with low access?

Cheers

Reviewer #2: The authors have done a great work, though it needs some corrections

Table 1. Descriptive data for the socioeconomic, demographic and area type variables separated by total, areas within and areas outside of the ‘golden hour'

Figures in the tables must be presented clearly, instead of median , better to present as number (%) for demographic & Socio-economic variables and Spatial variables

Grammatical errors should be rectified.

We appreciate the tools used in the manuscripts are acceptable and very well depicted.

Reviewer #3: The paper examines three variables presence of elderly, educational level and rural location. However, while it has provided multivariate analysis for first and third, it does not report the co-relation between elderly population and lower educational level. Expectedly, they would be co-related. In the review of literature, the authors do not elaborate on the reasons why these disparities would persist even in a highly developed and small country like Sweden. The authors also do draw any inferences from primary research on the human factors involved in the delay - are elderly anxious about hospitalisation? Do mobility issues affect their ability to alert neighbors, welfare staff who could call for the ambulance, in case they can not.

While it is acceptable that this paper only analyses data on infrastructure and broad demographic characteristics of the area, some effort is required to bring in more depth in the analysis and description by referring to other literature, which would help to complete the whole picture.

6. PLOS authors have the option to publish the peer review history of their article (what does this mean?). If published, this will include your full peer review and any attached files.

Reviewer #1: No

Reviewer #2: No

Reviewer #3: **Yes: **Neha Madhiwalla

---

## [Author Response · Author response to Decision Letter 0]

8 Nov 2021

Reviewer #1

1. Abstract, please add statistical analysis in brief in this section.

Response:

We agree that parts of the statistical analysis was missing in the abstract, and added a sentence about the ANOVA testing on page 2, row 35-38. 

2. The whole structure of your manuscript is not good and for example, the introduction and the methods sections are a bit messy. Some information regarding method reported in the introduction. Please report the method based on the stages of the study. Also the results based on the study questions.

Response:

We agree the structure could be improved. Several changes were made to improve the readability and flow of the paper. Using the research questions as reference, we created new headings both in ‘Methods’ and ‘Results’: 

1) The title “Methods and materials” was changed to “Data and methods”, page 12, row 268 and the subtitle “Pre-processing the data” was replaced with a subtitle called “Methods” because this was a part of carrying out the spatial analysis and not of the data description. 

2) Then, subtitles reflecting the research questions were added under it. These were “Analysis of spatial accessibility to EHC” which included the network analysis used to answer the first research question, page 13, row 290, “Assessment of differences in accessibility by measures” where the ANOVA testing used to answer question 2 is presented on page 14, row 334 and “Modelling disparities in accessibility” where the methodology for the modelling that answered the third research question is presented on page 15, row 347.

3) Similarly, we changed the titles in the Results section to reflect the order of the research questions, see subtitles on page 16, row 378 “Spatial patterns of accessibility to EHC”, on page 18, row 415 “Assessing differences in accessibility measures” and on page 19, row 431 “Modelling disparities in accessibility to EHC”. 

3. Please separate reference, for example references 1-5. Also, in my opinion the number of references is high and some references not closely related to your topic. for example references 2 and 4.

Response:

We reviewed the references and agree that the references should be separated. On page 3, row 52-54 we have now separated references that investigate urban and rural variations in response times, and references that investigate health outcomes and survival rates. While we agree that one reference was not closely related to these statements, we do think that the others were because they explicitly discuss the spatial variations in response times, and the effects it may have on patient health outcomes, which motivates our study. We also changed reduced the number of references from 5 to 2 on page 3, row 60. 

4. Please add the gap of evidence and also the novelty of your work in the introduction section.

Response:

We added a paragraph where the novelties, and the knowledge gap that this study fills, are presented on page 4, row 82-90. 

In relation to the second comment by reviewer 1, the description of the novelties is also aligned to the research questions of the paper to improve the flow of the paper. 

5. Provide a brief description of the emergency care system in the country in terms of statistics.

Response:

We added some basic information about the Swedish health care system, and some basic statistics of the prehospital emergency health care system, in the Study area section, page 10, row 219-228. 

6. In general, the article is very long. There is no need to review texts with this extent and it is better to summarize

Response:

Upon answering the reviewer comments, we have attempted to make adjustments to make the text more concise throughout the paper. 

7. Lines 178-9, on what basis are these suggestions made?

Response: 

We base these suggestions on the different types of medical interventions that different types of emergencies may require. A motivation to why three separate measures of accessibility are calculated was added in the Research design section, page 8, row 189-192 and page 9, row 196-199. 

8. The quality of figures is low. Please remove Fig 1

Response:

Figure 1 has now been removed, and the other figures have been renamed accordingly. 

9. In question 3, you stated some references. Explain the reason.

Response:

We deleted the references and reformulated the research questions. 

10. In my opinion you have 5 questions not 3.

Response:

See answer in previous point. We agree with the reviewer’s comments that the research questions were poorly formulated. These were changed to be more concise and to improve the flow of the article. The new questions, see page 9, row 210 - 213, refer to separate parts of the analysis.

11. The evidence related to "golden hour" is not enough at both national and international levels.

Response:

We appreciate the comment and agree that the ‘golden hour’ concept is not ideal. However, we find it useful for the purpose of comparing results with other international studies. We added two sentences where the concept is problematized in Methods section, page 14, row 327-332.

We have also added a paragraph in the Limitations section, page 26, row 577-582 where the use of this concept is discussed. For the purpose of this study we find it useful, although we acknowledge it could be less adequate to use in actual emergency health care planning. 

12. SEK? please add full term for the first time.

Response:

Thanks for pointing this out. We added “Swedish crowns” on page 16, row 385. 

13. Please report demographic information regarding table one at the first paragraph of the results section.

Response:

We report the urban and rural differences in TPT followed by some descriptive statistics of the demographic and socioeconomic characteristics in the first paragraph, page 12, row 383-387. 

14. Please separate the conclusion and the recommendations. IN conclusion you needs briefly address the answer to the aim/question of the study regardless of statistical difficulties. Also, remove recommendation from conclusion and report it under separate subheading. 

Response:

We agree the conclusion section was too broad. We rewrote it to increase readability and aligned the conclusions to the research questions. 

We also made a separate subheading called “Suggestions for future studies”, page 27, row 600, where we suggest futures studies. 

15. What is your practical recommendation for health policymaker and government bodies to decrease the number of state with low access?

Response:

They should use more than a measure of accessibility. While we analyze disparities in accessibility between places and groups, the main conclusion from this paper is that the definition of accessibility, i.e. how it is measured, influence the patterns that can be observed. As such, we added a paragraph in the Discussion section, page 25, row 550-556 where we recommend using not only RT as a measure of accessibility, as it risks obscuring places and groups with poor accessibility when it comes to accessing health care services at the hospital. Instead, TT and TPT should also be used to nuance the understanding of accessibility, and to show how accessibility may vary depending on the type of emergency and what medical interventions it requires. 

Reviewer #2

1. Table 1. Descriptive data for the socioeconomic, demographic and area type variables separated by total, areas within and areas outside of the ‘golden hour'. Figures in the tables must be presented clearly, instead of median , better to present as number (%) for demographic & Socio-economic variables and Spatial variables

Response:

We agree this makes interpretation easier and changed median values to indicate percentages instead of continuous numbers in Table 1. 

2. Grammatical errors should be rectified.

Response:

Thanks. 

3. We appreciate the tools used in the manuscripts are acceptable and very well depicted.

Response: 

Thanks. 

Reviewer #3

1. The paper examines three variables presence of elderly, educational level and rural location. However, while it has provided multivariate analysis for first and third, it does not report the co-relation between elderly population and lower educational level. Expectedly, they would be co-related.

Response:

The correlation between these two variables is low, r = 0,2. The rule of thumb is that we keep all the variables below r = 0,6. Now the bivariate correlation matrix is added as appendix. 

2. In the review of literature, the authors do not elaborate on the reasons why these disparities would persist even in a highly developed and small country like Sweden. 

Response:

In the Study area section, page 10, row 235-240, we note that reorganizations in health care in Sweden, like in other countries, has led to reduced accessibility to EHC for rural populations. We added information that Sweden spends a relatively high amount on health care in the EU context, page 10, row 221-224, although it is not an outlier in terms of spending. Combined with the fact that Sweden is a large country with relatively low population density, and that we measure accessibility as travel times, we would not expect that disparities would be lower in Sweden than elsewhere.

3. The authors also do draw any inferences from primary research on the human factors involved in the delay - are elderly anxious about hospitalisation? Do mobility issues affect their ability to alert neighbors, welfare staff who could call for the ambulance, in case they can not?

Response:

We agree that our analysis can be improved by adding literature that do not relate only to the spatial dimension of accessibility. We added a paragraph on other types of factors that influence levels of accessibility in the Theoretical background, page 7, row 167-172 to show that other factors than travel distance influence accessibility. We also added a paragraph in the Discussion section on page 25, row 539-545 where we briefly discuss that accessibility measured as travel times may not be accurate if other barriers hinder the patient from contacting the EHC in the first place, e.g. mental health status or negative perceptions based on previous encounters with the health care system. We also added a suggestion for future studies to include both spatial and aspatial factors simultaneously when studying accessibility on page 27-28, row 604-606.

4. While it is acceptable that this paper only analyses data on infrastructure and broad demographic characteristics of the area, some effort is required to bring in more depth in the analysis and description by referring to other literature, which would help to complete the whole picture.

Response:

Responding to the previous comments by reviewer 3, we think we have brought more depth to the analysis by putting the study into a broader context where also other types of factors are also discussed. Problematizing the use of ‘the golden hour’ also nuance the article and highlight the difficulties that this type of study entails in terms of assessing what is ‘good’ and what is ‘bad’ in terms of travel times.

---

## [Decision Letter · Decision Letter 1]

1 Dec 2021

Socio-spatial disparities in access to emergency health care - a Scandinavian case study

PONE-D-21-22117R1

Dear Dr. Hassler,

We’re pleased to inform you that your manuscript has been judged scientifically suitable for publication and will be formally accepted for publication once it meets all outstanding technical requirements.

Kind regards,

Srinivas Goli, Ph.D.

Academic Editor

PLOS ONE

Additional Editor Comments (optional):

Considering my own reading and reviewers suggestions, I am recommending this paper for publication in PLOS One.

Reviewers' comments:

Reviewer's Responses to Questions

**Comments to the Author**

1. If the authors have adequately addressed your comments raised in a previous round of review and you feel that this manuscript is now acceptable for publication, you may indicate that here to bypass the “Comments to the Author” section, enter your conflict of interest statement in the “Confidential to Editor” section, and submit your "Accept" recommendation.

Reviewer #1: All comments have been addressed

Reviewer #3: All comments have been addressed

2. Is the manuscript technically sound, and do the data support the conclusions?

Reviewer #1: Yes

Reviewer #3: Yes

3. Has the statistical analysis been performed appropriately and rigorously? 

Reviewer #1: Yes

Reviewer #3: I Don't Know

4. Have the authors made all data underlying the findings in their manuscript fully available?

Reviewer #1: Yes

Reviewer #3: Yes

5. Is the manuscript presented in an intelligible fashion and written in standard English?

Reviewer #1: Yes

Reviewer #3: Yes

6. Review Comments to the Author

Reviewer #1: Dear Respectable authors

Thank you for addressing comments. In my opinion, your manuscript is acceptable in this fashion.

Cheers

Reviewer #3: Within the constraints of the available data and their domain expertise, the paper makes a valuable contribution

7. PLOS authors have the option to publish the peer review history of their article (what does this mean?). If published, this will include your full peer review and any attached files.

Reviewer #1: **Yes: **Morteza Arab-Zozani

Reviewer #3: **Yes: **Neha Harshendra Madhiwalla

---

## [Editor Report · Acceptance letter]

3 Dec 2021

PONE-D-21-22117R1 

Socio-spatial disparities in access to emergency health care - a Scandinavian case study 

Dear Dr. Hassler:

I'm pleased to inform you that your manuscript has been deemed suitable for publication in PLOS ONE. Congratulations! Your manuscript is now with our production department. 

Kind regards, 

on behalf of

Dr. Srinivas Goli 

Academic Editor

PLOS ONE